# New Insights on Nucleotide Sequence Variants and mRNA Levels of Candidate Genes Assessing Resistance/Susceptibility to Mastitis in Holstein and Montbéliarde Dairy Cows

**DOI:** 10.3390/vetsci10010035

**Published:** 2023-01-03

**Authors:** Bothaina Essa, Mona Al-Sharif, Mohamed Abdo, Liana Fericean, Ahmed Ateya

**Affiliations:** 1Department of Animal Husbandry and Animal Wealth Development, Faculty of Veterinary Medicine, Damanhour University, Damanhour 22511, Egypt; 2Department of Biology, College of Science, University of Jeddah, Jeddah 21589, Saudi Arabia; 3Department of Animal Histology and Anatomy, School of Veterinary Medicine, Badr University in Cairo (BUC), Cairo 11829, Egypt; 4Department of Anatomy and Embryology, Faculty of Veterinary Medicine, University of Sadat, Sadat City 32897, Egypt; 5Department of Biology and Plant Protection, Faculty of Agricultural Sciences, University of Life Sciences King Michael I, 300645 Timisoara, Romania; 6Department of Animal Husbandry and Animal Wealth Development, Faculty of Veterinary Medicine, Mansoura University, Mansoura 35516, Egypt

**Keywords:** candidate gene, Holstein, Montbéliarde, gene expression, mastitis

## Abstract

**Simple Summary:**

Identification of markers to include in breeding plans is necessary in order to develop disease resistance to infectious diseases utilizing genetic control approaches. Recent studies used genome-wide association analysis to find new genes primarily responsible for the mastitis susceptibility of dairy cattle. Results, however, did not entirely persuade us that these genes were good candidates because we were unable to corroborate previously discovered SNP variants or genomic areas. In this study, SNPs linked to mastitis resistance/susceptibility were discovered in the *RASGRP1*, *NFkB*, *CHL1*, *MARCH3*, *PDGFD*, *MAST3*, *EPS15L1*, *C1QTNF3*, *CD46*, *COX18*, *NEURL1*, *PPIE*, and *PTX3* genes by PCR-DNA sequencing in Holstein and Montbéliarde cows with and without mastitis. The mRNA levels of these indicators also varied between healthy and affected dairy cows. Therefore, it may be useful to discover potential genes linked to mastitis susceptibility to improve the effectiveness of animal selection for innate resistance.

**Abstract:**

A major factor in the propagation of an infectious disease is host genetics. In this study, 180 dairy cows (90 of each breed: Holstein and Montbéliarde) were used. Each breed’s tested dairy cows were divided into two groups of comparable size (45 cows each), mastitis-free and mastitis-affected groups. Each cow’s jugular vein was punctured to obtain blood samples for DNA and RNA extraction. In the examined Holstein and Montbéliarde dairy cows, single nucleotide polymorphisms (SNPs) related with mastitis resistance/susceptibility were found in the *RASGRP1*, *NFkB*, *CHL1*, *MARCH3*, *PDGFD*, *MAST3*, *EPS15L1*, *C1QTNF3*, *CD46*, *COX18*, *NEURL1*, *PPIE*, and *PTX3* genes. Chi-square analysis of identified SNPs revealed a significant difference in gene frequency between mastitic and healthy cows. Except for *CHL1*, mastitic dairy cows of two breeds had considerably higher mRNA levels of the examined genes than did healthy ones. Marker-assisted selection and monitoring of dairy cows’ susceptibility to mastitis may be accomplished through the use of discovered SNPs and changes in the gene expression profile of the studied genes. These findings also point to a possible method for reducing mastitis in dairy cows through selective breeding of animals using genetic markers linked to an animal’s ability to resist infection.

## 1. Introduction

Mastitis is a typical infectious condition that affects dairy cows [1]. It has been demonstrated that it has an impact on farm productivity and animal welfare. Typically, it is described as a mammary gland inflammation caused by the entry and growth of pathogenic microorganisms [2]. It is a broad term for any mammary gland inflammation, ranging from mild inflammation that only increases milk’s somatic cell count to severe inflammation that results in gangrene and sepsis in one or more affected udder quarters [3]. This might also happen if the udder sustains heat, mechanical, or chemical damage. The first two months of lactation and the first 15 to 30 days postpartum are when females are most susceptible to developing mastitis [4,5]. As the cow’s number of lactations rises, it gets bigger [6].

Mastitis is the most expensive disease in the dairy industry [2,7]. Mastitis is still one of the most common diseases in dairy cattle despite major improvements in production features and negative genetic associations, particularly with milk output [8,9]. Loss of milk production, lowered milk quality, wasted milk, labour, veterinary care, culling due to mastitis, diagnostics, and preventative measures are all associated expenditures [8]. In order to avoid further financial losses, afflicted cows should be temporarily or possibly permanently withdrawn from milk production. According to recent studies, the average loss per affected cow is expected to be around $400 USD [10,11]. Furthermore, the estimated lactational incidence of mastitis for the first, second, and third or subsequent lactations was 0.35, 0.45, and 0.57, respectively [6].

Housing conditions, the epizootological environment in relation to the most prevalent causative infections, and the unique genetic makeup of each animal all have an impact on an animal’s susceptibility to mastitis [3]. The latter has long focused on breeding programs, although traditional selection techniques have had only sporadic success [12]. To fully understand a characteristic’s genetic architecture, it is crucial to identify genomic areas that have quantitative influence on that trait. These genomic regions can also be utilized to construct breeding strategies that would enhance the population’s frequency of favourable alleles [13]. This is especially crucial when features, such as disease resistance, have low heritabilities and are difficult to routinely record phenotypic. Although heritability for clinical mastitis appear to be fundamentally minor, being below 0.10 in the majority of investigations, genetic variability for immune system capacity appears to be noteworthy [14,15,16]. Both management techniques and the selection of mastitis-resistant genotypes are traditional ways to lower the incidence of mastitis in a herd. Quantitative trait loci (QTLs) linked to mastitis features have been discovered as a result of recent technical developments in cow genomics, such as candidate gene approach [13,17]. Mastitis incidence can be reduced with the help of genetic marker-assisted selection for mastitis traits because it produces more consistency and phenotypic discrimination than traditional selection [18].

It is possible to see resistance to mastitis as a complicated feature that is probably controlled by many genes with minor effects rather than a few genes with substantial effects [19]. Genome wide association analysis was used in previous studies to identify various genomic areas that contain potential mastitis susceptibility genes [20,21,22,23,24,25]. These investigations have been helpful in locating genetic variations linked to mastitis. However, they lack consistency in reporting single nucleotide polymorphisms (SNPs) implicated in susceptibility. This study’s main objective was to investigate the relationship between SNPs in candidate genes (*RASGRP1*, *NFkB*, *CHL1*, *MARCH3*, *PDGFD*, *MAST3*, *EPS15L1*, *C1QTNF3*, *CD46*, *COX18*, *NEURL1*, *PPIE*, and *PTX3*) and the incidence of mastitis resistance/susceptibility in Holstein and Montbéliarde dairy cows using PCR-DNA sequencing and real time PCR approaches.

## 2. Material and Methods

### 2.1. Ethics Statement

The Institutional Committee of the Faculty of Veterinary Medicine, Damanhour University, Egypt authorized the sample collection and animal care techniques utilized in this study (code DMU-VetMed- 2021/055).

### 2.2. Animals and Experimental Samples

A total of 180 dairy cows, 90 of each breed (Holstein and Montbéliarde), were employed in this investigation. Animals shared the same environment and fitted to the similar private farm in Ismailia Desert Road, Ismailia Governorate, Egypt. The test was conducted from January 2022 to April 2022. The cows were grown in a commercial dairy herd of about 450 animals, which was in its third lactation season. Cows normally weighed 450 kg and were 4.5 years old. The animals were kept in a cubicle (free-stall/feedlot) barn with straw-bedded stalls, a slatted floor that was scraped frequently, a total mixed ration (TMR), twice-daily milking, and artificial insemination. The dairy cows under investigation had complete clinical examinations in accordance with the recommended methods. Based on the prevalence of mastitis and the animals’ general health, 45 dairy cows from each breed were divided into two groups of equal size. The first group, which was designated as the mastitis-free group, had cows that were clinically healthy (have a history of mastitis resistance in previous lactations, meaning mastitis was never noticed in those lactations) The second group, which included cows exhibiting mastitis, was known as the group with mastitis (excessive body heat, insufficient appetite, swollen and painful udder, reddish and yellowish milk colour and foul odour, clotted milk, teat cracks). The California mastitis test was also used to determine whether the investigational cows had mastitis on a regular basis. Veterinarians and their skilled assistants examined the animals for mastitis while farm workers kept an eye on them constantly. All mammary gland quarters were thoroughly examined visually and physically during clinical veterinary examinations.

Each cow in each group had its jugular vein punctured to obtain five millilitre of blood. To extract DNA and RNA, blood samples were obtained into an EDTA-anticoagulant-contained vacutainer. At 20 °C, blood samples were maintained frozen until DNA extraction. Freshly drawn blood samples were submitted right away for RNA extraction to prevent RNA hydrolysis.

### 2.3. DNA Extraction and Polymerase Chain Reaction (PCR)

All blood samples were subjected to DNA extraction using the commercial kit QIAamp DNA Mini kit and blood (Qiagen, Hilden, Germany), in accordance with the established methodology (DNA purification from blood or body fluids). Nanodrop’s DNA extraction and quantification software (NanoDrop Technologies, Wilmington, DE, USA). Only samples with A260/A280 ratios between 1.7 and 1.9 and DNA concentrations between 5 and 40 ng/µL were deemed appropriate for analysis.

The *RASGRP1*, *NFkB*, *CHL1*, *MARCH3*, *PDGFD*, *MAST3*, *EPS15L1*, *C1QTNF3*, *CD46*, *COX18*, *NEURL1*, *PPIE*, and *PTX3* genes’ coding regions were amplified using PCR. The primer sequences were created using the *Bos taurus* sequence that was published in PubMed. Table 1 provides a list of the primers utilized in the amplification.

In a thermal cycler, the polymerase chain reaction mixture was run in a final volume of 150 μL. Each reaction volume comprised 1.5 μL of each primer, 75 μL of PCR master mix (Jena Bioscience, Germany), 6 μL of DNA, and 66 μL of d.d. water. The first denaturation temperature of 95 °C was applied to the reaction mixture for 4 min. The process of cycling included 35 cycles of denaturation at 95 °C for one minute, annealing at temperatures (as stated in Table 1) for one minute, extension at 72 °C for one minute, and a final extension at 72 °C for ten minutes. At 4 °C, samples were kept. A gel documentation system was used to view fragment patterns under UV after representative PCR results were found using agarose gel electrophoresis.

### 2.4. DNA Sequencing and Polymorphism Detection

Before DNA sequencing, non-specific bands, primer dimmers and other contaminants were removed using a PCR purification kits (Jena Bioscience # pp-201s/Munich, Hamburg, Germany). Therefore, the target PCR product of expected size will be obtained [26]. Nanodrop (Uv-Vis spectrophotometer Q5000, Waltham, MA, USA) was used for PCR quantification to obtain high quality and concentrations [27]. PCR products containing the target band were sent for DNA sequencing in both the forward and reverse directions in order to find SNPs in healthy and mastitis-affected dairy cows. These products were sequenced with an ABI 3730XL DNA sequencer (Applied Biosystems, Waltham, MA, USA) and the enzymatic chain terminator technique developed by Sanger et al. [28].

Chromas 1.45 and BLAST 2.0 Softwares were used to analyse DNA sequencing data [29]. SNPs were identified as differences between the PCR products of the investigated genes and GenBank reference sequences. The MEGA4 software was used to identify differences in the amino acid sequence of the inspected genes among enrolled dairy cows based on a sequence alignment [30].

### 2.5. Total RNA Extraction, Reverse Transcription and Quantitative Real-Time PCR

Following the manufacturer’s instructions, Trizol reagent was used to extract total RNA from the blood of the dairy cows under investigation (RNeasy Mini Ki, Catalogue No. 74104). Using a NanoDrop^®^ ND-1000 Spectrophotometer, the isolated RNA’s quantity was determined and validated. Each sample’s cDNA was created in accordance with the production methodology (Thermo Fisher, Waltham, MA, USA, Catalog No, EP0441). Using quantitative RT-PCR and SYBR Green PCR Master Mix, the expression patterns of the genes *RASGRP1*, *NFkB*, *CHL1*, *MARCH3*, *PDGFD*, *MAST3*, *EPS15L1*, *C1QTNF3*, *CD46*, *COX18*, *NEURL1*, *PPIE*, and *PTX3* were evaluated (2× SensiFastTM SYBR, Bioline, CAT No: Bio-98002). Real-time PCR utilising SYBR Green PCR Master Mix was used to relative quantify the quantity of mRNA (Quantitect SYBR green PCR kit, Toronto, ON, Canada, Catalog No, 204141). As shown in Table 2, primer sequences were created using the *Bos taurus* sequence that was published in PubMed. As a constitutive control, *ß. actin* gene was used for normalization. An amount of 25 µL of total RNA, 4 µL of Trans Amp buffer, 0.25 µL of reverse transcriptase, 0.5 µL of each primer, 12.5 µL of Quantitect SYBR green PCR master mix, and 8.25 µL RNase-free water made up the reaction mixture. The finished reaction mixture was put in a thermal cycler and subjected to the following programme: reverse transcription at 50 °C for 30 min, initial denaturation at 94 °C for 8 min, followed by 40 cycles of 94 °C for 15 s, annealing temperatures as stated in Table 2, and 72 °C for 30 s. After the amplification phase, a melting curve analysis was performed to confirm the specificity of the PCR product. The 2^−ΔΔCt^ approach was used for exploring the comparative expression of each gene in the tested sample in proportion to *ß. actin* gene [31,32].

### 2.6. Statistical Analysis

H_0_: Nucleotide sequence variants and mRNA levels of *RASGRP1*, *NFkB*, *CHL1*, *MARCH3*, *PDGFD*, *MAST3*, *EPS15L1*, *C1QTNF3*, *CD46*, *COX18*, *NEURL1*, *PPIE*, and *PTX3* genes could not assess resistance/susceptibility to mastitis in dairy cows of Holstein and Montbéliarde breeds.

H_A_: Nucleotide sequence variants and mRNA levels of *RASGRP1*, *NFkB*, *CHL1*, *MARCH3*, *PDGFD*, *MAST3*, *EPS15L1*, *C1QTNF3*, *CD46*, *COX18*, *NEURL1*, *PPIE*, and *PTX3* genes assess resistance/susceptibility to mastitis in dairy cows of Holstein and Montbéliarde breeds.

Chi-square analysis was used for determining the significant differences in discovered SNPs of genes between the 180 dairy cows. For this purpose, statistical analysis was performed using Graphpad statistical software program (Graphpad prism for Windows version 5.1, Graphpad software, Inc., San Diego, CA, USA). When *p* < 0.05 or *p* < 0.01 was reached, a difference was significant or highly significant, respectively. Using the SPSS programme version 23 (one-way analysis of variance test and multiple comparison Tukey’s HSD test were used) was used to compare the means of mRNA levels for the researched groups in the investigated genes analysed. Statistical parameters were expressed as mean ± standard error (SE). Using the gene expression profile of the researched genes as an independent variable, a discriminant analysis model was utilized to evaluate the relevance of many variables in order to distinguish between affected and healthy dairy cows as a dependent variable. The goal was to discriminate between mastitic and healthy cows relied on the mRNA levels of genes under investigation. The interaction between two factors (gene type and mastitis resistance/susceptibility) and its impact on the gene expression outcomes parameter was evaluated using a univariate general linear model (GLM) with two-way ANOVA.

## 3. Results

### 3.1. Nucleotide Sequence Variants of Investigated Genes

In the examined Holstein and Montbéliarde dairy cows, nucleotide sequence variations in the form of SNPs kinked to mastitis resistance/susceptibility were detected in the *RASGRP1* (410-bp), *NFkB* (396-bp), *CHL1* (547-bp), *MARCH3* (455-bp), *PDGFD* (531-bp), *MAST3* (650-bp), *EPS15L1* (383-bp), *C1QTNF3* (526-bp), *CD46* (288-bp), *COX18* (511-bp), *NEURL1* (476-bp), *PPIE* (324-bp), and PTX3 (617-bp) genes. By comparing the nucleotide sequence differences between the examined genes and the reference sequences provided in GenBank, all identified SNPs were validated (Appendix A). Chi-square analysis of the identified SNPs revealed that the frequencies of the investigated genes were highly substantially (*p* < 0.0001) different between the unaffected and affected Holstein and Montbéliarde dairy cows (Table 3). The variants identified in Table 3 are all located within exonic region of studied genes; resulting in coding mutations between mastitis healthy and affected dairy cows.

### 3.2. Gene Expression Pattern of Investigated Markers

Figure 1 displays the expression profile of the examined markers. Compared to healthy ones, mastitic Holstein and Montbéliarde dairy cows, levels of *RASGRP1*, *NFkB*, *MARCH3*, *PDGFD*, *MAST3*, *EPS15L1*, *C1QTNF3*, *CD46*, *COX18*, *NEURL1*, *PPIE*, and *PTX3* gene expression were significantly higher. The *CHL1* gene, meanwhile, experienced considerable down-regulation.

The kind of gene and mastitis resistance/susceptibility in each breed significantly influenced the mRNA levels of the examined indicators. The greatest potential levels of mRNA were found for the mastitis-affected dairy cows’ *NFkB* (2.63 ± 0.11) and *COX18* (2.64 ± 0.16) genes, respectively, while *CHL1* (0.44 ± 0.12 and 0.64 ± 0.11) had the lowest levels in both Holstein and Montbéliarde dairy cows. In the same way, *CHL1* was found to have the highest possible levels of mRNA among all genes tested (1.78 ± 0.18 and 2.36 ± 0.11) in healthy dairy cows of Holstein and Montbéliarde breeds, respectively. While RASGRP1 had the greatest values (0.42 ± 0.06) in healthy Montbéliarde, CD46 was found to have the lowest levels (0.48 ± 0.14) in Holstein.

## 4. Discussion

Mastitis is a complicated disorder, and numerous functional candidate genes may play a role in its onset, susceptibility, as well as recovery [19]. Numerous genetic variations that can either improve or impair health and productivity are present in the genetic composition of farm animal species. These variations include different types of single nucleotide polymorphisms (SNPs) [33]. The majority of cases are brought on by SNPs that influence gene function to varied degrees, such as switching one amino acid for another, duplications and deletions that induce premature translation termination and frame shift, and complete deletion of whole exon(s) or gene(s) in affected individuals. It is understood that these changes to the coding regions have an impact on the behaviour of mRNA splicing patterns or protein function [34].

For the purposes of this investigation, we carried out PCR-DNA sequencing for the genes *RASGRP1*, *NFkB*, *CHL1*, *MARCH3*, *PDGFD*, *MAST3*, *EPS15L1*, *C1QTNF3*, *CD46*, *COX18*, *NEURL1*, *PPIE*, and *PTX3*. Between mastitis-affected Holstein and Montbéliarde dairy cows and healthy control cows, SNPs (single nucleotide polymorphisms) were found. A highly significant distribution (*p* < 0.0001) in the detected SNPs was discovered using chi-square analysis. It is significant to highlight that, the polymorphisms found and published here disclose new information about the studied genes when compared to the corresponding GenBank reference sequence.

Recent studies did the genome wide association analysis to target new genes specific for mastitis susceptibility in cattle [20,21,22,23,24,25]. However, until now, no research has looked at these genes’ SNPs and their association with mastitis susceptibility. Our study is the first to demonstrate this association using bovine (*Bos taurus*) gene sequences published in PubMed. To our knowledge, no studies have looked into the polymorphism of the genes *RASGRP1*, *NFkB*, *CHL1*, *MARCH3*, *PDGFD*, *MAST3*, *EPS15L1*, *C1QTNF3*, *CD46*, *COX18*, *NEURL1*, *PPIE*, and *PTX3* and its relatedness to mastitis in cattle

Monitoring the health of mastitic animals was performed using the candidate gene method [35,36,37,38]. In contrast to earlier studies, this study investigated polymorphisms via SNP genetic markers to compare the prevalence of mastitis in two breeds of dairy cow (Holstein and Montbéliarde). Genetic characterization of breeds, biodiversity evaluation, and conservation decisions have all been transformed by the SNP genetic marker [39]. SNP research may offer a more accurate understanding of the evolution of European cattle than other markers [40,41]. SNPs are also thought to be particularly important in the search for connections between a marker at an unidentified gene locus and a known site in the genome. It is possible to assess a phenotypic effect by understanding its genetic basis, making the search for such relationships essential [42,43].

Transcript abundance functions as a heritable endophenotype and is associated with chromosomal polymorphisms, in accordance with the genetic genomics theory [44]. This method supported the idea that combining data on gene expression and chromosomal variants could aid in our understanding of the genetics underlying the onset of disease [45]. Quantitative trait loci (QTLs) are polymorphisms connected to gene expression [46]. In the current study, we proposed that mastitis susceptibility transcriptional response individual genetic variation may affect the course of the disease.

The mRNA levels of the genes *RASGRP1*, *NFkB*, *CHL1*, *MARCH3*, *PDGFD*, *MAST3*, *EPS15L1*, *C1QTNF3*, *CD46*, *COX18*, *NEURL1*, *PPIE*, and *PTX3* were measured using real-time PCR in resistant and susceptible Holstein and Montbéliarde dairy cows. Our research showed that mastitic dairy cows had higher levels of *RASGRP1*, *NFkB*, *MARCH3*, *PDGFD*, *MAST3*, *EPS15L1*, *C1QTNF3*, *CD46*, *COX18*, *NEURL1*, *PPIE*, and *PTX3* gene expression than resistant dairy cows. For the *CHL1* gene, though, the tendency was the opposite. Our study is the first to apply a real-time PCR technique for identifying the mRNA levels of these markers in resistant and susceptible dairy cows to mastitis.

Previous research used genetic markers such as RFLP and SNP to analyse the polymorphism of immune genes and their relationship to ruminant susceptibility to mastitis [35,36,37,38]. To address the shortcomings of earlier studies, we examined gene polymorphism using gene expression and SNP genetic markers. As a result, the mechanisms of the investigated gene regulation are well understood in both mastitic and healthy dairy cows. We are aware of few data on the gene expression profile of indicators related with mastitis susceptibility in cattle. The expression of antioxidant genes in milk from cows with clinical mastitis caused by Staphylococcus aureus and Escherichia coli was compared in a study by Asadpour et al. [47], it was shown that *SOD* expression was significantly up-regulated. Additionally, *GPx* was significantly overexpressed in mastitis milk caused by *E. coli* as compared to S. aureus in terms of mRNA levels. According to Darwish et al. [48], sheep with postpartum problems had considerably lower amounts of mRNA for the *SOD* and *CAT* genes than did resistant ewes.

The RAS guanyl releasing protein 1 (*RASGRP1*) gene controls T cell receptor signalling as well as lymphocyte activation, development, and function [49]. Pathogen challenge causes *RASGRP1* to express differently, suggesting a potential role in ruminant mastitis [50]. The typical macrophages activated by S. aureus were shown to have nuclear factor kappa B (NFkB) [51]. By invading the macrophages, the bacterium begins the process of activating NFkB, which then initiates the production of pro-inflammatory cytokines and the following inflammatory response. NFkB activation causes inflammatory responses in bovine mammary epithelial cells, according to Boutet et al. [52]. Cell adhesion molecule L1 like (CHL1) is a member of the L1 family, is controlled by stress levels, affects immune system functions, and is involved in cell migration and the prevention of neuronal cell death [53]. In depressive patients with persistent stress, monocyte *CHL1* expression was markedly downregulated [54]. Additionally, there were less positive CD19+ and CD20+ B cells in these patients. The immune system suffers, and disease vulnerability is raised when these two immune cells are downregulated [54]. Based on the aforementioned findings, we hypothesize that an increase in *CHL1* may have a favourable impact on the immune system and, thus, on the health of the udder.

E3 ubiquitin-protein ligase MARCH3 is also known as membrane-associated ring-CH-type finger (*MARCH3*). It produces an E3 ubiquitin protein ligase enzyme that has been associated with a number of biological functions, such as regulating the endosomal transport route and membrane trafficking [55]. MAST3 (microtubule associated serine/threonine kinase (3) has been shown to play a role in the maturation of lymphocytes. The function of *MAST3* genes in the processing of antigens has been reported. *MAST3* was found to be highly expressed in lymphocytes and antigen-presenting cells in a gene expression experiment [56]. Additionally, it has been demonstrated that *MAST3* knockdown reduced the level of Toll-like receptor-4-dependent NF-kappaB, which is required for the innate immune response to be triggered during pathogen invasion [56]. According to reports, platelet-derived growth factor D (PDGFD) has a role in macrophage recruitment and inflammatory modulation [57]. As a result, the genes *MARCH3*, *MAST3*, and *PDGFD* were proposed as potential causative factors for mastitis susceptibility.

The gene EPS15L1 (epidermal growth factor receptor pathway substrate 15 like 1) has been identified as being essential for the development of T lymphocytes in Zebrafish [58]. Since the gene is largely conserved among the proteins produced by Zebrafish, mice, and humans, it is anticipated to perform similarly in other mammals, such as cattle [58]. The membrane assault complex includes complement components, which are crucial for immunological response, antibody formation, inflammation, and phagocytosis of bacterial cells [59]. During the postpartum period, these were found to be related to mastitis [4]. Tumour necrosis factor related protein 3 (*C1QTNF3*), another complement, was discovered to be related to somatic cell score [17]. Similar findings were made regarding the SNP in the *CD46* gene and mastitis in cows [60].

The *COX18* gene codes for a mitochondrial cytochrome c oxidase assembly component that is necessary for the insertion of integral membrane proteins into the mitochondrial inner membrane. Additionally, it is necessary for cytochrome c oxidase assembly and function [61]. The neutralized E3 ubiquitin protein ligase 1 (*NEURL1*) gene on BTA26 has a known SNP that is located in intron 1 and has been linked to mastitis risk in cattle [20]. In Nordic cow breeds, *NEURL1* has also been linked to fat content [62]. The peptidylprolyl isomerase E (*PPIE*) gene has been linked to the adaptive immune system and is believed to play a role in protein folding [63]. The immune system in mammals uses comparable ways to control cellular inflammation, thus it is probable that the *PPIE* gene also causes the immune system in cattle to become activated. The innate immune response to intra-amniotic infection and inflammation may involve the gene pentraxin 3 (*PTX3*) [64]. This protein is expressed by different mesenchymal and epithelial cell types, especially endothelial cells and mononuclear phagocytes, in response to inflammatory stimuli. Staphylococcus aureus, one of the main causes of mastitis, was discovered to cause an upregulation of *PTX3* [65]. This gene’s function in innate immunity and inflammatory control has been mentioned in a number of research studies [65,66]. According to a review of gene-targeted mice and genetic correlations in people, *PTX3* is essential for resistance to a variety of infections, including Escherichia coli [67]. As a result, we propose that this gene may be a candidate for mastitis susceptibility.

Bacterial infections, particularly those caused by Escherichia coli, Streptococcus uberis, and Staphylococcus aureus, are a major cause of mammary gland inflammation [68]. As a result, the cows are exposed to a greater number of pathogens, which stimulates their immune system. A network of mastitis pathways controlled neutrophil and other leukocyte activity during inflammation of the mammary gland. Achieving a balance between pathogen eradication and excessive tissue damage appears to be particularly dependent on the prompt and carefully controlled movement of leukocytes to infection loci [69]. When macrophages and epithelial cells are exposed to either lipoteichoic acid (LTA, from Gram-positive bacteria) or LPS (from Gram-negative bacteria), which both promote the release of TNF and IL1B, the neutrophil recruitment cascade is started. During an infection, complement C5a levels are also elevated, which causes mast cells to produce histamine [59]. Vascular endothelial cells respond when TNF, IL1, C5a, and histamine interact with their specific receptors [70]. Emigrated neutrophils ascend a new chemotactic gradient before migrating through the epithelium into the mammary gland lumen, where they are essential for the eradication of pathogens [71,72]. Our hypothesis is that dairy cows have crucial, shared innate immune defence mechanisms against various intra-mammary infection sources, and that variations in the critical genes linked to these defence mechanisms can result in variances in disease resistance.

The substantial shift of the expression pattern of immune markers (*RASGRP1*, *NFkB*, *CHL1*, *MARCH3*, *PDGFD*, *MAST3*, *C1QTNF3*, *CD46*, *COX18*, *NEURL1*, *PPIE*, and *PTX3*) in mastitic cows may be caused by the activity of phagocytic cells for secretion of the proinflammatory cytokines and the subsequent inflammation that affects the harmed tissue [49,50]. Therefore, we assume that the majority of the mastitis cases in this study that impacted dairy cows were caused by an infectious agent. Additionally, our Real Time PCR results offer strong evidence that the mastitic cows were undergoing a severe inflammatory response.

## 5. Conclusions

By PCR-DNA sequencing of the *RASGRP1*, *NFkB*, *CHL1*, *MARCH3*, *PDGFD*, *MAST3*, *EPS15L1*, *C1QTNF3*, *CD46*, *COX18*, *NEURL1*, *PPIE*, and *PTX3* genes, single nucleotide polymorphisms (SNPs) associated with mastitis resistance/susceptibility were found between mastitis healthy and affected Holstein and Montbéliarde. In order to provide additional information and enable genomic region prioritization, the current study provided the association of SNP markers with mastitis incidence. Additionally, these indicators’ mRNA levels changed between healthy and affected dairy cows. These distinctive functional variants offer a promising opportunity to lessen cow mastitis through selective breeding of animals employing genetic markers connected to natural resistance. Variable gene expression profiles in dairy cows that are resistant and susceptible to mastitis may act as a guide and a biomarker for assessing health. Future mastitis therapy may be made easier by the gene targets revealed here.

## Figures and Tables

**Figure 1 vetsci-10-00035-f001:**
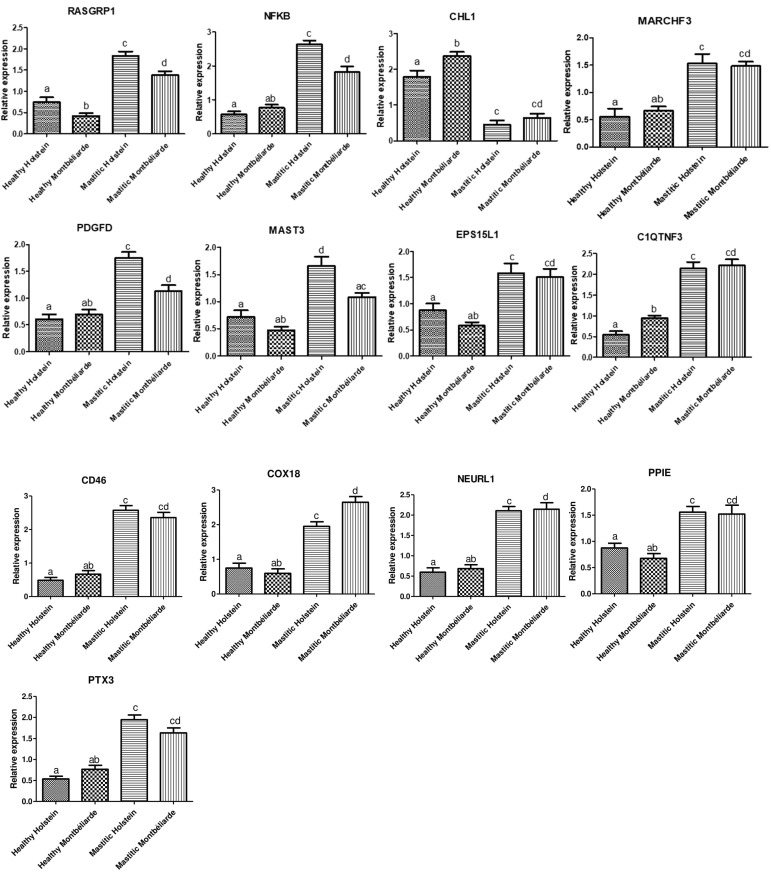
Comparative expression patterns of the genes *RASGRP1*, *NFkB*, *CHL1*, *MARCH3*, *PDGFD*, *MAST3*, *EPS15L1*, *C1QTNF3*, *CD46*, *COX18*, *NEURL1*, *PPIE*, and *PTX3* in healthy and mastitis affected Holstein and Montbéliarde dairy cows. a, b, c, d, ab, cd means small alphabetic letters show significance when *p* < 0.05.

**Table 1 vetsci-10-00035-t001:** Oligonucleotide primers sequence investigated genes utilized in PCR-DNA sequencing.

Gene	Forward	Reverse	Annealing Temperature (°C)	Length of PCR Product (bp)	Reference
*RASGRP1*	5′- ACTCATGGCTGCAGAGCAGTC-3′	5′- TCCATGGTGCTTGCCAGGCTG-3′	62	410	Current study
*NFKB*	5′- TCAGGTCAAACTCCAGAATGGC-3′	5′- GCCATTCTGGAGTTTGACCTGA-3′	58	396	Current study
*CHL1*	5′- CGTGCAGATCGGCTGGGAGCT-3′	5′- ATTGTTAGATACAATTATCCGA-3′	60	547	Current study
*MARCHF3*	5′- CTCTACGCGGCTGTCCGCCTC-3′	5′- GGTCCGCACTACTGTTGACAG-3′	64	455	Current study
*PDGFD*	5′- GCCAGCGAGTGCGGGCGCGCGT-3′	5′- GTAGTCATCAGACTTGAACGT-3′	62	531	Current study
*MAST3*	5′- TCCTGTTACCGCTCCTTACCCA-3′	5′- CTTGTGGTTCAACAGAGAGTGAG-3′	62	650	Current study
*EPS15L1*	5′- TCCATTATATGAGTCTTACTA-3′	5′- AACCATTGACAGGTAAGAGGCT-3′	58	383	Current study
*C1QTNF3*	5′- CGAGGAGACCACGGCGGCCAGA-3′	5′- CCGGTCATGACATCAAAGAAGT-3′	60	526	Current study
*CD46*	5′- CCGCTGAAGGCGCCGCTCCGC-3′	5′- ACAGCCCTCCTGGAGAGACGAC-3′	62	288	Current study
*COX18*	5′- TGCGAGCGCGCGTGGTCTGTGA-3′	5′- TAGGTAAGTGAGCCTGGCAAC-3′	64	511	Current study
*NEURL1*	5′- GTAACAACTTCTCCAGTATTC-3′	5′- CTCCTCAGGCAGCGCCTTGGCC-3′	58	476	Current study
*PPIE*	5′- GCAAGAGCAAGATGGCCACTAC-3′	5′- ACTTCTTCAACCAGTCATCATC-3′	60	324	Current study
*PTX3*	5′- TCCAGCAATGCATATCTCTGTGA-3′	5′- TCATTGGTGTCACCGGATGCAC-3′	62	617	Current study

*RASGRP1* = RAS guanyl releasing protein 1; *NFKB* = Nuclear factor kappa B subunit; *CHL1* = Cell adhesion molecule L1; *MARCHF3* = Membrane associated ring-CH-type finger 3; *PDGFD* = Platelet derived growth factor D; *MAST3* = Microtubule associated serine/threonine kinase 3; *EPS15L1* = Epidermal growth factor receptor pathway substrate 15 like 1; *C1QTNF3*= C1q and TNF related 3; *CD46* = Cluster of differentiation 46; *COX18* = Cytochrome c oxidase assembly factor; *NEURL1* = Neuralized E3 ubiquitin protein ligase 1, *PPIE* = Peptidylprolyl isomerase E, and *PTX3* = Pentraxin 3.

**Table 2 vetsci-10-00035-t002:** Oligonucleotide primers sequence of investigated genes used in real time PCR.

Gene	Primer	Product Length (bp)	Annealing Temperature (°C)	Accession Number	Source
*RASGRP1*	F5′-GAGAAGCTCCACGGAAACCA-3′ R5′-CAGAGGCACCATCATTCGGA-3′	137	60	NM_001144078.1	Current study
*NFKB*	F5-CAGATGGGCTACACTGAGGC-3′ R5′-TGCGGAAGGAGGTCTCTACA-3′	184	60	NM_001076409.1	Current study
*CHL1*	F5′-CGGTTTCCTCGAAGGAAGGT-3′ R5′-GAAGGAGGCAGCCCAGAAAG-3′	172	59	NM_001205541.3	Current study
*MARCHF3*	F5′-TGGAGACATGGTGTGCTTCC-3′ R5′-TCGAGCCGACTGCTAAAGTG-3′	105	58	NM_001077941.1	Current study
*PDGFD*	F5′-GGCTCTCGTTGACATCCAGT-3′ R5′-GTAAGTTCGGTTGCTGGTGG-3′	167	62	NM_001083706.1	Current study
*MAST3*	F5′-CCTTACCCAGACTGGAGTGTC-3′ R5-CAGCCTCCTGCAGCAAATG-3′	211	60	XM_024994781.1	Current study
*EPS15L1*	F5′-GAGTTCTCTGCCTTCCGTGC-3′ R5′-GGTGATGGTGTGAGGTTCCG-3′	144	59	XM_024993963.1	Current study
*C1QTNF3*	F5′-ATAGAGCTCTGTTGACTGGCCG-3′ R5′-ACTCCATGCCAGTGTGTGTAA-3′	119	59	NM_001101138.1	Current study
*CD46*	F5′-AGTTAGTGGCACACACTGGG-3′ R5′-CCACGTGCCTTACCCAAGAT-3′	161	60	NM_001242563.2	Current study
*COX18*	F5′-ATGCGGAGGCTTGTTTCAGA-3′ R5′-CGGAGAGCGACAGACATGAA-3′	113	60	NM_001082437.2	Current study
*NEURL1*	F5′-GGTAACAACTTCTCCAGTATTCCCA-3′ R5′-TTGTGGTGGCATCGGTGAGA-3′	131	58	NM_001192253.3	Current study
*PPIE*	F5-CTGACGTGTGACAAGACGGA-3′ R5′-TCCCCACAGTCGGAGATGAT-3′	149	59	NM_001098161.1	Current study
*PTX3*	F5-GAACGTCGTCTCTCCAGCAA-3′ R5′-TGTCCCACTCGGAGTTCTCA-3′	191	60	NM_001076259.2	Current study
*ß. actin*	F5′-GCTCAGAGCAAGAGAGGCAT-3′ R5′-CACACGGAGCTCGTTGTAGA-3′	117	60	AF191490.1	Current study

*RASGRP1* = RAS guanyl releasing protein 1; *NFKB* = Nuclear factor kappa B subunit; *CHL1* = Cell adhesion molecule L1; *MARCHF3* = Membrane associated ring-CH-type finger 3; *PDGFD* = Platelet derived growth factor D; *MAST3* = Microtubule associated serine/threonine kinase 3; *EPS15L1* = Epidermal growth factor receptor pathway substrate 15 like 1; *C1QTNF3* = C1q and TNF related 3; *CD46* = Cluster of differentiation 46; *COX18* = Cytochrome c oxidase assembly factor; *NEURL1* = Neuralized E3 ubiquitin protein ligase 1, *PPIE* = Peptidylprolyl isomerase E, and *PTX3* = Pentraxin 3.

**Table 3 vetsci-10-00035-t003:** SNP distribution and kind of mutation for the genes under investigation in in healthy and mastitic Holstein and Montbéliarde dairy cows.

Gene	SNPs	Healthy *n* = 90		Mastitic *n* = 90		Total	Type of Mutation	Amino Acid Number and Type	Chi Value	*p* Value
		Holstein *n* = 45	Montbéliarde *n* = 45	Holstein *n* = 45	Montbéliarde *n* = 45					
*RASGRP1*	C99T	24	-	-	-	24/180	Synonymous	33 F	20.02	<0.0001
	T276C	-	31	-	-	31/180		92 V	25.86	
*NFKB*	C213A	-	-	34	-	34/180	Synonymous	71 P	28.36	
*CHL1*	A117T	-	36	-	-	36/180	Synonymous	39 G	30.03	
*MARCHF3*	C86T	-	23	-	-	23/180	Non-synonymous	29 S to L	19.19	
	T116G	-	29	-	-	29/180		39 F to C	24.19	
	G216A	-		22	15	37/180	Synonymous	72 S	30.86	
*PDGFD*	G94A	-		13	13-	13/180	Non-synonymous	32 V to I	10.84	
	A140G	-39		-	-	39/180		47 D to G	32.53	
	G232C	29-			-	29/180		78 E to Q	24.19	
	C303A	19 33			-	52/180	Synonymous	101 T	43.38	
*MAST3*	T47C	-26			-	26/180	Non-synonymous Non-synonymous	16 V to A	21.69	
	A155G	-		37	-	37/180		52 D to G	30.86	
	A384C	28-			-	28/180	Synonymous	128 I	23.36	
*EPS15L1*	C34T	23	11	-		34/180	Non-synonymous	12 P to S	28.36	
	T79C		-	19-		19/180		27 S to P	15.85	
	A202G	-	26	-		26/180		68 T to A	21.69	
	A250G	-		31-		31/180		84 T to A	25.86	
	T280G	-	23	-		23/180		94 S to A	19.18	
*C1QTNF3*	G41A		-	-21		21/180	Non-synonymous	14 R to Q	17.52	
*CD46*	C27T	17	-	-		17/180	Synonymous	9 P	14.18	
	G217A		-	28-		28/180	Non-synonymous	73 V to I	23.36	
	C243T	34	29	-		63/180	Synonymous	81 L	52.55	
	G131T		-	36-		36/180		44 R to L	30.03	
	G272A	-	18	-		18/180		91 R to H	15.02	
	T339C	-	33	-		33/180		113 G	27.53	
	A466C		-	27-		27/180		156 T to P	22.52	
*NEURL1*	G56A		-	21-		21/180	Non-synonymous	19 R to H	17.51	
	C108A	-	19	-		19/180	Synonymous	36 S	15.85	
	C269T	28	-	-		28/180	Non-synonymous	90 T to M	23.36	
*PPIE*	T80C	-		31-		31/180	Non-synonymous	27 M to T	25.86	
	T134C		-	29-		29/180	Non-synonymous	45 M to T	24.19	
	T287C	-	15	-		15/180	Non-synonymous	96 L to P	12.51	
*PTX3*	A106G	-	37	-		37/180	Non-synonymous	36 *n* to D	30.86	
	C189T	-		21		21/180	Synonymous	63 H	17.51	
	C364G	-		18		18/180	Non-synonymous	122 P to A	15.01	
	C488A		32	-		32/180	Non-synonymous	163 A to E	26.69	

*RASGRP1* = RAS guanyl releasing protein 1; *NFKB* = Nuclear factor kappa B subunit; *CHL1* = Cell adhesion molecule L1; *MARCHF3* = Membrane associated ring-CH-type finger 3; *PDGFD* = Platelet derived growth factor D; *MAST3* = Microtubule associated serine/threonine kinase 3; *EPS15L1* = Epidermal growth factor receptor pathway substrate 15 like 1; *C1QTNF3* = C1q and TNF related 3; *CD46* = Cluster of differentiation 46; *COX18* = Cytochrome c oxidase assembly factor; *NEURL1* = Neuralized E3 ubiquitin protein ligase 1, *PPIE* = Peptidylprolyl isomerase E, and *PTX3* = Pentraxin 3. A = Alanine; C = Cisteine; D = Aspartic acid; E = Glutamic acid; F = Phenylalanine; G = Glycine; H = Histidine; I = Isoleucine; L = Leucine; M = Methionine; P = Proline; Q = Glutamine; R = Arginine; S = Serine; T = Threonine; and V = Valine.

## Data Availability

On reasonable request, the corresponding author will provide the data that underpin the study’s conclusions.

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
