# Peer review of "New Insights on Nucleotide Sequence Variants and mRNA Levels of Candidate Genes Assessing Resistance/Susceptibility to Mastitis in Holstein and Montbéliarde Dairy Cows"

_vetsci, 2023, doi:10.3390/vetsci10010035_

Round 1

Reviewer 1 Report

1 this study was to examine the association of SNP in candidate genes (RASGRP1, NFkB, CHL1, MARCH3, PDGFD, MAST3,  EPS15L1, C1QTNF3, CD46, COX18, NEURL1, PPIE, and PTX3) influencing the incidence of mastitis resistance/ susceptibility in Holstein and Montbéliarde dairy cows using PCR- DNA sequencing and real time PCR approaches.The article has some innovation, but the experimental content is less, and there are some problems that need to be modified.

1.All tables in the manuscript should be three line tables

2. The description of statistical analysis is too complicated. Please simplify it. Each statistical analysis method can be described in the labes to the specific results of the figures and tables.

3. Figure 1 is too unclear to obtain effective information

Author Response

Comment

1 this study was to examine the association of SNP in candidate genes (RASGRP1, NFkB, CHL1, MARCH3, PDGFD, MAST3,  EPS15L1, C1QTNF3, CD46, COX18, NEURL1, PPIE, and PTX3) influencing the incidence of mastitis resistance/ susceptibility in Holstein and Montbéliarde dairy cows using PCR- DNA sequencing and real time PCR approaches. The article has some innovation, but the experimental content is less, and there are some problems that need to be modified.

Response

  • We thank reviewer for this positive comment. In fact our manuscript provides new insights on nucleotide sequence variants and mRNA levels of candidate genes assessing resistance/susceptibility to mastitis in Holstein and Montbéliarde dairy cows.
  • In recent investigations, genome-wide association analysis was utilized to identify new genes specifically for dairy cattle's susceptibility to mastitis. Results, however, did not entirely persuade us that these genes were good candidates because we were unable to corroborate previously discovered SNP variants or genomic areas. In this study, SNPs linked to mastitis resistance/susceptibility were discovered.

Comment

  1. All tables in the manuscript should be three line tables

Response

We are grateful to the reviewer for drawing it to our consideration. All tables in the manuscript are changed to three line tables.

  1. The description of statistical analysis is too complicated. Please simplify it. Each statistical analysis method can be described in the labes to the specific results of the figures and tables.

Response

We are grateful to the reviewer for drawing it to our consideration. Statistical analysis is simplified.

Comment

  1. Figure 1 is too unclear to obtain effective information

Response

We thank reviewer for this comment. Better quality figure is added.

Reviewer 2 Report

 General comments

I believe that the direction of the undertaken research is valuable for application reasons. It makes it possible to eliminate mastistis by means of selection methods.

The authors refer to as many as 74 literature items, which may suggest that the undertaken research direction was the subject of numerous studies, or that the article is of a review nature. I think that both the Introduction and the Discussion chapters could be written more synthetically (I leave this only to the authors' consideration).

It is necessary to provide a more detailed description of the statistical methods used, e.g. discrimination analysis. How was the model built?

I propose to supplement table 3 with an additional column containing the probability (P).

Figure number 1 (p. 10) is hard to read.

Detailed comments

Line 105-106: Perhaps I don't understand something? How is it possible that the third lactation cows were 3 years old?

Lines 202 and 206: Replace Ho and HA with H0 and HA.

Line 209: The phrase "Statistical parameters" is incorrect.Controlled traits were characterized statistically by calculating mean ± standard deviation (SD)”.

Line 211: Replace ANOVA with “analysis of variance”.

Line 212: P £ 0.05 = significant, P £ 0.01 = highly significant

Lines: 217-219: How was it possible to study interactions using univariate analysis?

Lines 220-221: Remove the sentence "Date were expressed as ..."

Lines 280-281: The article is missing probability levels. Therefore, it is difficult to refer to the statement contained in the lines.

Lines 329-330: What are the names of the genes listed for?

Line 419: “Montbéliarde In order” - character "." is missing

Author Response

General comments

Comment

I believe that the direction of the undertaken research is valuable for application reasons. It makes it possible to eliminate mastitis by means of selection methods.

Response

We thank reviewer for this positive comment. Our findings point to a possible method for reducing mastitis in dairy cows through selective breeding of animals using genetic markers linked to an animal's ability to resist infection.

Comment

The authors refer to as many as 74 literature items, which may suggest that the undertaken research direction was the subject of numerous studies, or that the article is of a review nature. I think that both the Introduction and the Discussion chapters could be written more synthetically (I leave this only to the authors' consideration).

Response

  • We thank reviewer for leaving this comment to the authors' consideration.
  • In fact, the introduction section contains information about mastitis and its implications on animal, utilization of marker assisted selection and candidate gene approach for selection of natural resistant animal. Finally, the introduction contains at its last section a novelty statement for doing our study as in recent investigations, genome-wide association analysis was utilized to identify new genes specifically for dairy cattle's susceptibility to mastitis. Results, however, did not entirely persuade us that these genes were good candidates because we were unable to corroborate previously discovered SNP variants or genomic areas. In this study, SNPs linked to mastitis resistance/susceptibility were discovered.
  • In the same respect, discussion section contains deciphering for our results. Additionally discussion contains information about the investigated genes for justification of utilization of these genes in our study particularly no studies have looked into the polymorphism of the genes RASGRP1, NFkB, CHL1, MARCH3, PDGFD, MAST3, EPS15L1, C1QTNF3, CD46, COX18, NEURL1, PPIE, and PTX3 and its relatedness to mastitis in cattle.
  • For the aforementioned reasons and behalf of authors, I see all cited references and data in introduction and discussion sections are necessary to be mentioned.

Comment

It is necessary to provide a more detailed description of the statistical methods used, e.g. discrimination analysis. How was the model built?

Response

We are grateful to the reviewer for drawing it to our consideration. A more detailed description of the statistical methods is mentioned.

Comment

I propose to supplement table 3 with an additional column containing the probability (P).

Response

We are grateful to the reviewer for drawing it to our consideration. An additional column containing the probability (P) is added to table 3.

Comment

Figure number 1 (p. 10) is hard to read.

Response

We thank reviewer for this comment. Better quality figure is added.

Detailed comments

Comment

Line 105-106: Perhaps I don't understand something? How is it possible that the third lactation cows were 3 years old?

Response

We thank reviewer for this comment. We apologize for this miswriting type. It is corrected.

Comment

Lines 202 and 206: Replace Ho and HA with H0 and HA.

Response

We thank reviewer for this comment. Ho and HA is replaced with H0 and HA.

Comment

Line 209: The phrase "Statistical parameters" is incorrect. “Controlled traits were characterized statistically by calculating mean ± standard deviation (SD)”.

Response

We thank reviewer for this positive comment. The statistical analysis section is illustrated.

Comment

Line 211: Replace ANOVA with “analysis of variance”.

Response

We thank reviewer for this comment. ANOVA is replaced with with “analysis of variance”.

Comment

Line 212: P £ 0.05 = significant, P £ 0.01 = highly significant

Response

We thank reviewer for this comment. Words are added.

Comment

Lines: 217-219: How was it possible to study interactions using univariate analysis?

Response

We thank reviewer for this comment. The statistical analysis section is illustrated.

Comment

Lines 220-221: Remove the sentence "Date were expressed as ..."

Response

We thank reviewer for this comment. The sentence "Date were expressed as ..." is removed

Comment

Lines 280-281: The article is missing probability levels. Therefore, it is difficult to refer to the statement contained in the lines.

Response

We thank reviewer for this comment. Probability levels are added.

Comment

Lines 329-330: What are the names of the genes listed for?

Response

We are grateful to the reviewer for drawing it to our consideration. We apologize for this miswriting type. We have deleted the names of the genes.

Comment

Line 419: “Montbéliarde In order” - character "." is missing

Response

We are deeply indebted to the reviewer for this comment. The character is added.

Round 2

Reviewer 1 Report

I have no other questions